# Rescue Procedures after Suboptimal Deep Brain Stimulation Outcomes in Common Movement Disorders

**DOI:** 10.3390/brainsci6040046

**Published:** 2016-10-08

**Authors:** Adam M. Nagy, Christopher M. Tolleson

**Affiliations:** Department of Neurology, Vanderbilt University Medical Center, 1161 21st Avenue South, A-0118 Medical Center North, Nashville, TN 37232, USA; adam.nagy@vanderbilt.edu

**Keywords:** Parkinson’s disease, essential tremor, dystonia, deep brain stimulation, treatment failure, rescue leads

## Abstract

Deep brain stimulation (DBS) is a unique, functional neurosurgical therapy indicated for medication refractory movement disorders as well as some psychiatric diseases. Multicontact electrodes are placed in “deep” structures within the brain with targets varying depending on the surgical indication. An implanted programmable pulse generator supplies the electrodes with a chronic, high frequency electrical current that clinically mimics the effects of ablative lesioning techniques. DBS’s efficacy has been well established for its movement disorder indications (Parkinson’s disease, essential tremor, and dystonia). However, clinical outcomes are sometimes suboptimal, even in the absence of common, potentially reversible complications such as hardware complications, infection, poor electrode placement, and poor programming parameters. This review highlights some of the rescue procedures that have been explored in suboptimal DBS cases for Parkinson’s disease, essential tremor, and dystonia. To date, the data is limited and difficult to generalize, but a large majority of published reports demonstrate positive results. The decision to proceed with such treatments should be made on a case by case basis. Larger studies are needed to clearly establish the benefit of rescue procedures and to establish for which patient populations they may be most appropriate.

## 1. Introduction

Deep brain stimulation (DBS) is a unique and exciting functional neurosurgical therapy, allowing for easily adjustable post-surgical changes to an implanted programmable device, which maximizes long-term clinical outcomes. It involves neurosurgical implantation of multicontact electrodes either unilaterally or bilaterally in specific anatomical areas “deep” within the brain. Targets vary depending on the indication for surgery. These electrodes are connected by a tunneled extension wire to a pulse generator, or neurostimulator, typically implanted in a subcutaneous pocket below the clavicle. The neurostimulator provides a modifiable, high frequency electrical current that modulates the neurocircuitry surrounding the electrodes, clinically mimicking the effects of ablative stereotactic lesioning techniques [1,2]. The stimulation parameters (amplitude, pulse width and frequency) as well as activated contact(s) can be easily adjusted by the treating physician in the clinic. Depending on the comfort level of the physician, patients can also be given varying levels of control over their programming settings. DBS has Food and Drug Administration (FDA) approval in two conditions, Parkinson’s disease (PD) and essential tremor (ET). DBS also has a humanitarian device exemption for dystonia and obsessive compulsive disorder. Outside of these approved indications, DBS has been used for a variety of conditions including Parkinson’s plus disorders, Huntington’s disease, Tourette’s, Alzheimer’s, epilepsy, chronic pain, major depression, post-traumatic stress disorder, and schizophrenia among others [3]. In general, DBS is indicated in all its approved conditions when prospective patients are symptomatically severe and refractory to medical treatment, or when medication adverse effects become intolerable [4].

The process of selecting proper candidates for DBS, successfully implanting the electrode(s) in the proper location, and appropriately programming the DBS device is complex. When done correctly, the efficacy of DBS is well established [5,6,7,8]. However, in some instances, even under optimal circumstances with a multidisciplinary team approach, DBS can lead to suboptimal results either immediately after the initial surgery or, later, as symptom benefit declines [9,10,11]. When common and reversible complications such as hardware issues, improper lead positioning and inadequate programming are ruled out, the patient and clinicians are left with the difficult decision of what to do next. DBS centers have trialed various options including the use of additional or “rescue” DBS lead(s), moving the established lead to another location, or subsequently using lesioning therapy. Given a paucity of data, it can be difficult for centers to make decisions. It can also be difficult to counsel patients and their families as outcomes are not clear. Obtaining consent and appropriately detailing out risks and benefits in such situations comes with inherent difficulties and should be given careful attention by the DBS team [12,13]. This review will highlight the available data on some of the techniques used after suboptimal DBS results in PD, ET, and dystonia patients.

## 2. Parkinson’s Disease

Parkinson’s disease (PD) is a neurodegenerative disorder characterized by the progressive loss of dopaminergic neurons and the buildup of intracellular inclusions called Lewy bodies composed of alpha synuclein. Clinically, it manifests as both the diagnostic motor features (tremor, bradykinesia, rigidity, and postural instability) as well as a myriad of non-motor features (autonomic instability, neuropsychiatric decline, sleep disorders, pain, etc.). PD medications primarily involve modulation of the dopaminergic pathway and target motor features, whereas non-motor features are much more difficult to treat.

DBS is a proven adjunctive surgical therapy for treatment of the motor symptoms of PD [6,14,15]. It is currently approved for levodopa-responsive PD patients with at least 4 years of disease not adequately controlled with medication or whose treatment is complicated by medication-related side effects (i.e., motor fluctuations, dyskinesias). The surgery has been approved for PD since 2002 with recent indications suggesting that earlier usage in the disease may be effective [16]. Compared to best medical therapy alone, DBS in conjunction with medication has been proven far superior. Studies have demonstrated a notable improvement in quality of life, motor scores, and a reduction of wearing off in patients who have received DBS [17,18,19]. Overall benefits have been maintained for up to 11 years according to long-term follow-up studies [20,21,22] although there is some concern that any initial benefit in gait or posture may deteriorate more quickly [23].

The most commonly used targets are the subthalamic nucleus (STN) and globus pallidus internus (GPi). Large comparative trials have demonstrated equal benefit in regards to the overall treatment of PD motor symptoms between targets [6,24]. However, there were subtle differences identified, which are still continuously being explored. For example, STN stimulation classically has allowed for greater medication reduction post-surgery, and GPi stimulation has seemed more advantageous for patients with depression, greater balance difficulty, and impairments in verbal fluency [6,24,25]. The longest follow-up in the large trials directly comparing the two targets was 24 months [6,24]. There is still no unifying consensus on specific criteria for favoring one target over the other with the current data. Different DBS centers still differ on their approaches for target selection and often will use the target with which they have the most experience and comfort.

DBS in these two targets can be said to have “failed” for many reasons. Some of the more common reasons include improper patient selection, suboptimal electrode placement, suboptimal management (programming and medications), hardware complications (infection, lead fracture, dead battery), and progression of PD such that symptoms not modified by DBS become the patient’s primary disability [11]. Exclusive of these more common reasons for failure of PD DBS, patients can still have inadequate or progressively lessening motor benefit despite good lead positioning and programming parameters [10,11]. This can manifest as reemergence of dystonia, worsening in motor fluctuations and dyskinesias, and progression in the cardinal motor symptoms (tremor, bradykinesia, rigidity) initially modified by DBS [10,11]. In certain patients, up titration of dopaminergics in attempts to adequately control breakthrough symptoms can also lead to disabling side effects (neuropsychiatric changes, fatigue, sleep disruption, impulse control disorders, orthostatic hypotension, and dyskinesia among others). Case studies and series have reported using additional leads in the other primary PD DBS target (i.e., STN stimulation to rescue failed GPi stimulation and vice versa) in these circumstances with some success [15,26,27,28,29,30].

Published cases where patients underwent GPi stimulation for failed STN stimulation generally report patients with a young age of disease onset (average age 41) and a long interval of success with their initial STN surgery (average of 8 years until rescue surgery) [26,27,29,30]. In all reviewed cases, the reason for failure was disabling dyskinesia or dystonia. Discontinuation of STN stimulation after initiation of GPi stimulation sometimes led to worse control of cardinal motor features, leading to continued STN stimulation in four of seven cases reviewed [26,27,29,30]. Of the seven cases, six achieved a clinically significant benefit with the addition of GPi stimulation [26,27,29,30]. Benefits included reduction in the dystonia, dyskinesia, and levodopa equivalent daily dose (i.e., total medication needed) [26,27,29,30].

In the six cases reviewed where STN stimulation was used to rescue failed GPi stimulation, the reason for failure was worsening motor symptoms soon after surgery (within the first 2–3 years) [15,28]. Average age at disease onset was also in the 40s (exact ages at onset unavailable from one study). Notably, in three of the six cases, there were hardware complications that necessitated the removal of at least one GPi electrode prior to the use of rescue leads [15,28]. In all cases, STN stimulation replaced GPi stimulation, rather than stimulation of both regions, as the GPi leads were already removed in all patients [15,28]. STN stimulation led to improved United Parkinson’s disease Rating Scale Part III (UPDRS-III) scores, decreased levodopa daily dose, and cessation of or improvement in dyskinesia [15,28].

Another interesting rescue procedure reported by Deligny et al. was that of a bilateral subthalamotomy performed through DBS electrodes prior to their removal due to hardware infection [31]. In this case, radiofrequency subthalamotomy through the leads led to a durable benefit in measured motor scores, dyskinesia and off times [31]. Unfortunately, after the procedure, there were some mild cognitive and motivation side effects witnessed as well [31].

Finally, other targets besides the STN and GPi have been explored in PD DBS such as the pedunculopontine nucleus (PPN). This target has primarily been used for PD patients suffering from freezing of gait (FOG) and other gait disturbances, either as the sole target or in combination with STN DBS. It has the potential to be used as a rescue target in the future but further study is needed. The PPN was initially chosen as a potential DBS target given work in animal models which has shown the PPN plays a significant role in the normal operation of axial muscles which help regulate posture and gait [32,33,34]. FOG is a disabling symptom commonly seen in PD where patients literally “freeze” to the floor when they attempt to ambulate. It commonly occurs with the initiation of gait, with turning or when maneuvering in tight spaces such as doorways and crowds [35,36]. FOG is generally refractory to medications and STN/GPi DBS [37]. Several studies were reviewed that looked at the PPN as the sole target in PD patients with postural instability and gait symptoms (PIGD). A recent meta-analysis was performed on 10 such studies [37]. While there was a statistically significant improvement in motor symptoms and postural instability, the meta-analysis did not find a significant improvement in FOG [37]. The improvement in motor symptoms was also less substantial with PPN stimulation than has been found with STN or GPi stimulation [37]. In contrast, an interesting study by Stefani et al. looked at six PD patients with axial signs simultaneously implanted in the PPN and STN [38]. Patients were analyzed 2–6 months after surgery in ON/OFF medication states with either or both targets activated. The PPN was particularly effective for gait and posture, and the combination of the targets being “on” was superior to one alone [38]. Liu et al. also report a case of a PD patient with FOG implanted simultaneously with bilateral PPN and STN leads [39]. The investigators did not test the leads in both targets simultaneously due to problems with dizziness when all leads were activated. However, on testing individual targets, sole stimulation of the PPN leads did show some benefit in the gait problems whereas the STN leads did not [39].

In summary, case reports of rescue procedures for PD have been generally positive, though follow-up has been short. Typically, rescue procedures have involved stimulation of the approved target that was not originally implanted (i.e., STN for failed GPi or vice versa). Subthalamotomy performed through existing DBS electrodes has also been attempted. Rescue procedures seem to be more common in those with a young age of disease onset. Stimulation of the PPN is also an intriguing idea both as an initial therapy and a rescue therapy for those with more axial symptoms and FOG, but currently further study is still needed.

## 3. Essential Tremor

Essential tremor (ET) is the most commonly seen movement disorder [40]. Clinically, it typically manifests as a bilateral action and postural tremor. The disease commonly runs in families, suggesting a hereditary component, yet a specific genetic cause has not been identified [40]. Common therapeutic medications include primidone, beta blockers, topiramate and gabapentin.

DBS is a proven surgical option for medication refractory ET. The accepted target is the ventral intermediate (VIM) nucleus of the thalamus. Unilateral DBS for ET received FDA approval in 1997 although it is commonly used bilaterally. Response to stimulation is often robust, with studies demonstrating >80% tremor improvement [7,8]. Studies have also demonstrated that this benefit can be persistent over a long period of time (>12 years) [41,42,43].

In some cases, however, the benefits of DBS either are suboptimal or diminish irrespective of inaccurate lead placement, hardware complications or other device issues [11,44]. Possible reasons for diminishing response include disease progression and tolerance to stimulation. Several different rescue techniques have been attempted to improve tremor control in these patients. Some DBS centers have attempted to either add additional leads or reposition in a second target. These other targets have primarily included the STN, caudal zona inserta (cZI), or the ventralis oralis anterior nucleus of the thalamus (VOA) [9,45,46]. Many of these newer targets were selected to see if stimulation of the subthalamic areas (STN, cZI, prelemniscal radiations) would produce similar results to previously used lesioning approaches. Subthalamotomy has been an efficacious surgical option for tremor dating back to the 1960s [47,48,49]. Our institution has also tried adding a second thalamic lead anterior to the VIM, using the combination of the VOA and VIM leads together to direct current away from structures causing stimulation-induced side effects and allow for more aggressive stimulation parameters [50,51,52].

In a case series by Blomstedt et al., patients who had failed VIM stimulation underwent re-implantation in the cZI [9]. In this series, they reported a 57% improvement in tremor control with cZI stimulation, compared to a 25% improvement in tremor control with prior VIM stimulation [9]. Within their cohort (*n* = 5), however, two of the patients had relatively immediate failure of their VIM leads (<6 months) [9]. This may suggest an initial improper positioning of the leads as opposed to true superiority of the cZI over VIM. The other three patients received their cZI lead implantation an average of 9 years after VIM implantation with benefit [9].

In two series by Mehanna et al. and Oyama et al., a total of seven patients underwent a second operation into the VOA, STN, or prelemniscal radiations for failed VIM stimulation [45,46]. Six of the seven patients had mild to moderate improvement after reoperation [45,46]. In some patients, the physicians used simultaneous stimulation of both the new target and VIM, while others had stimulation of the new target alone [45,46]. A shortcoming of these series is the heterogeneity of the patient population and treatment strategies. Four of the seven patients did not have ET [45,46]. Rather, two had multiple sclerosis induced tremor, one had tremor from a treated thalamic anteriovenous malformation and one had an atypical tremor of unknown cause [45,46]. In the three patients with ET, one had maximal tremor control with the VOA rescue lead stimulated alone. The other two had maximal tremor control with simultaneous stimulation of VIM and the new target (one in the VOA and one in the prelemniscal radiations). The varying treatment strategies (different targets and different combinations of stimulation) make this data difficult to interpret.

Yu et al., Isaacs et al. and Sukul et al. from our institution [50,51,52] published a series in which they used the placement of a second electrode in the thalamus antero-medially to the VIM to direct stimulation away from structures causing stimulation-induced side effects (such as the internal capsule and ventralis caudalis nucleus of the thalamus) [50,51,52]. Limiting side effects from the initial lead included severe paresthesias, diplopia, dysarthria, and dizziness. The leads were connected in parallel to a common voltage source allowing more control over the field of stimulation [50,51,52]. Directing the stimulation away from unintended targets allowed for more aggressive stimulation parameters with essentially equivalent or better tremor control and reduced side effects in all patients [50,51,52].

Bahgat et al. retrospectively reviewed seven patients with ET who underwent unilateral thalamotomy as a rescue procedure after failed VIM DBS [53]. Reasons for failure in these patients included intolerable side effects, malpositioned electrodes, and symptom progression [53]. After thalamotomy, six of the seven patients reported symptomatic improvement, though only three of those six reported corresponding functional improvement and one patient reported no improvement at all [53]. However, only one patient had a significant persistent adverse effect in the form of facial numbness from thalamotomy after DBS [53].

In summary, case reports of rescue procedures for ET have also been generally positive, though cohorts have been notably heterogeneous. Several newer targets have been tried in combination with and as a replacement for VIM stimulation, which has raised the possibility of a synergistic effect of stimulating different regions. Thalamotomy for failed DBS had modest success. An alternative rescue approach with favorable results has been implanting additional thalamic leads to direct stimulation away from structures responsible for intolerable side effects.

## 4. Dystonia

Dystonia is an unusual movement disorder characterized by sustained and repetitive muscle contraction, often resulting in abnormal posturing [54]. The exact pathophysiology is not known, but the origin appears to be in the basal ganglia. Dystonia can be from a variety of causes such as genetic abnormalities, neurodegenerative conditions, structural changes or insults to the brain, chemical exposures, or medications among others [55,56]. It also has a varied clinical presentation, presenting either as a focal dystonia (i.e., isolated to one body part) versus a more generalized or segmental dystonia [55,56]. Common non-surgical treatments for dystonia include botulinum toxin injections, anticholinergic medications, and benzodiazepines.

DBS is a proven surgical treatment for dystonia refractory to medication and botulinum toxin injections, particularly primary generalized dystonia (i.e., genetic or idiopathic) [5]. This treatment received Humanitarian Device Exemption from the FDA in 2003. The primary target has been the GPi in the majority of cases due to prior experience with lesioning therapies and the use of the target in the PD population [57]. More recent DBS cases have used the STN target as well, most commonly with focal cervical dystonia [5,58]. Some studies report up to a 60%–70% improvement on dystonia rating scales in generalized dystonia post GPi DBS [5,58,59,60,61]. Results in focal dystonias are more variable. The most common dystonia, cervical dystonia, does tend to have good response post DBS [62,63,64]. The responses to DBS in other types of focal or segmental dystonia are less well defined and reported less in the literature [5]. Post-surgical programming in this population can also often be challenging compared to PD and ET. Unlike these movement disorders where stimulation results in immediate clinical results, there is a long latency between programming adjustments and resulting clinical benefit that can be months in duration.

Like the other movement disorders already discussed, dystonia patients can have a suboptimal response to DBS despite good lead positioning and a lack of detectable reversible complications such as hardware malfunction or poor programming parameters [46,65]. Various techniques have been applied in the dystonia population. A case series by Oyama et al. reported two patients who underwent rescue lead placement for dystonia [46]. The first was a patient with cervical dystonia who had incomplete benefit from bilateral GPi stimulation and underwent implantation of a second rescue lead into the left GPi [46]. The rationale was that the patient’s original left GPi lead was 2.4 mm more anterior than the right on repeat imaging. The second was a tardive dyskinesia/dystonia patient who had incomplete benefit from bilateral GPi stimulation and underwent implantation of bilateral STN rescue leads two years later [46]. Both patients had complete symptom resolution with stimulation of both the original and rescue leads [46]. In the first case, the authors stimulated all contacts immediately after the addition of the third lead with excellent results. In the second case, the authors attempted to reduce GPi stimulation in favor of purely STN stimulation but only with activation of all four leads did the patient achieve maximum benefit. Benefit was sustained at 17 months for the left GPi rescue operation and 15 months for the bilateral STN rescue procedure at the time of publication [46].

No other case series were found that specifically addressed rescue leads for suboptimal DBS results in dystonia. There was one report where bilateral STN DBS was used as a rescue procedure for a failed unilateral pallidotomy [65]. Also, Schjerling et al. did perform a study directly comparing the STN to the GPi as targets for dystonia [66]. Part of the study did address simultaneous stimulation of the two targets. The study was a randomized, double-blind crossover study, and all patients received both STN and GPi leads [66]. The study included 13 patients and was quite heterogeneous; ages ranged from 12 to 57 years, disease duration ranged from 3 to 30 years, and it was roughly half generalized and half focal dystonia [66]. While the results did not demonstrate a statistically significant difference between GPi and STN stimulation, there was a trend toward greatest improvement with simultaneous stimulation of both targets, followed by STN alone and then GPi alone [66]. As with the single case report mentioned above, there may be a role for simultaneous stimulation of the STN and GPi targets in patients who have failed to achieve beneficial results with a single target.

In summary, there is little available data regarding rescue lead implantation for dystonia patients. In the one case report reviewed in addition to the study employing simultaneous GPi and STN stimulation, the increased benefit of combined STN and GPi stimulation is intriguing and could be looked at more intensely in additional studies with larger populations. The published number of cases is currently extremely limited. Studies of DBS in dystonia are complicated by the variety of clinical presentations and underlying causes of dystonia. Studies are also complicated by the delay in benefit from stimulation which can take several months to manifest, if not longer. This makes programming inherently difficult as well as determining what qualifies as a DBS failure.

## 5. Discussion and Conclusions

As described, published rescue procedures for failed DBS in PD, ET, and dystonia have been performed with generally positive reported results and do have a role in cases of suboptimal DBS outcomes. However, the data is still quite difficult to apply to any general population for several reasons. There is a high degree of individualization that takes place between institutions and patients when it comes to such procedures. The amount of data is still very limited, and is currently entirely in the form of case reports and series. Further, there is little incentive to write or publish case reports of negative outcomes after rescue procedures, likely creating significant publication bias.

Ultimately, more established guidelines, utilizing more concrete data, are needed for performing rescue therapies in suboptimal DBS outcomes in each of the indications discussed. However, a consensus set of guidelines based on better data from prospective blinded, randomized clinical trials may be difficult to achieve given the ethics of performing a blinded randomized trial of a rescue surgical therapy as well as the multiple differing variables present in situations where rescue leads are required. Nonetheless, with the uncertainty of the current data and the difficulty of putting a patient through another surgery with poorly established outcomes, it is a given that rescue procedures should only be performed as a last resort, after every attempt to optimize the current DBS lead has been undertaken. Postoperative imaging should be performed to verify proper lead placement, hardware checked to make sure its functioning appropriately, medications optimized to the fullest, and programming adjustments exhausted as much as possible. There should be multidisciplinary discussions between experienced DBS neurologists and neurosurgeons before undertaking these procedures where targets and procedural options are discussed and weighed. The risks, ethics, and potential emotional distress of putting patients and their families through another brain surgery should never be taken lightly.

This review summarizes the available data on rescue therapies post DBS. More information and experience from DBS centers, both good and bad, is needed to better establish future guidelines and techniques. Still, available data does suggest that some patients can achieve benefits with rescue procedures. The decision to proceed with such treatment should be undertaken with caution and involve open discussions with a team of DBS physicians, patients and their families, fully explaining the uncertainty of results.

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
