# Peer review of "Rescue Procedures after Suboptimal Deep Brain Stimulation Outcomes in Common Movement Disorders"

_brainsci, 2016, doi:10.3390/brainsci6040046_

Round 1

Reviewer 1 Report

In the last years there has been a plenty of reviews devoted to DBS. Most of these papers are ripetitive and do not offer any critical or innovative aspects that may justify their appearance in the literature. This review is a further example of what is happening, offering few new elements and considering once again aspects and concepts more or less already known.

- The Authors still consider traditional DBS targets in a way that is outdated, not taking into account the feasibility of a taylored surgery according to symptoms. No mention of the utility of DBS of the CM-Pf in MD has been provided.

 - As far as DBS of the PPN is concerned, the Authors refer to the pioneering paper by Stefani et al. while ignoring the former paper  of literature ( Mazzone et al 2005 Neuroreport) both the successive and more important advancing in PPN-DBS  realized by Mazzone et al. and the innovative concepts arising from those studies (for example in  J Neural Transm  2016;123:751-67; Neurosurgery. 2013, 73:894-906; J Neural Transm  2011, 118:1431-51 etc..............).

- The lack of originality characterize also the sections devoted to ET and dystonia, where the Authors in an attempt to be all-encompassing create confusion and the reader does not receive adequate information on what to do and what are the best indications for DBS. This is particular evident when they in an attempt to correlate correct targeting and clinical outcome  base their thought on old lesional concepts, without any reference to a more actual view of DBS functioning that considers the possibility of an action carried out on neuronal pathways rather than on a single neuronal  population.

- The discussion is very  short  and does not fit well  into the merits of the matter described.

I- n conclusion, the review is not acceptable as it stands, and requires to be profoundly revised according to more actual concepts that consider DBS acting on neuronal pathways rather on single neuronal populations, so that the whole output of a neuronal network is modulated to an extent as close as possible to a physiological normal state.

Author Response

We appreciate the reviewer taking the time to review our manuscript.  We have attempted to address comments below.

In the last years there has been a plenty of reviews devoted to DBS. Most of these papers are ripetitive and do not offer any critical or innovative aspects that may justify their appearance in the literature. This review is a further example of what is happening, offering few new elements and considering once again aspects and concepts more or less already known.

We would respectfully disagree with the reviewer.  This was an invited review asking for a review on clinical applications of DBS hence the topic.  Our topic and outline were cleared by the editor.  While we would agree there are many general reviews of DBS, our point was to focus on rescue therapies for the main movement disorder indications so as not to repeat information already discussed in the literature at large which was the understandable concern of the reviewer.  There is very little in the literature on rescue therapies other than case reports which is discussed in the body of our paper.

- The Authors still consider traditional DBS targets in a way that is outdated, not taking into account the feasibility of a taylored surgery according to symptoms. No mention of the utility of DBS of the CM-Pf in MD has been provided.

The requested review was for clinical applications of DBS.  Our focus was on rescue therapies after suboptimal DBS outcomes and not tailoring initial surgeries to specific symptoms.  While we strongly agree tailoring initial therapy to symptoms is extremely important, initial DBS therapy was not the focus of this review.  As stated in the introduction, we were also focusing on purely the movement disorders currently approved for DBS (eg PD, ET, and dystonia) and not other diseases for which DBS has been used such as MD.

- As far as DBS of the PPN is concerned, the Authors refer to the pioneering paper by Stefani et al. while ignoring the former paper of literature ( Mazzone et al 2005 Neuroreport) both the successive and more important advancing in PPN-DBS realized by Mazzone et al. and the innovative concepts arising from those studies (for example in J Neural Transm 2016;123:751-67; Neurosurgery. 2013, 73:894-906; J Neural Transm 2011, 118:1431-51 etc..............).

We realize the contributions of Mazzone et al and have now cited his 2005 paper as rightfully requested.  Our goal however was not to articulate the underlying concepts behind PPN stimulation but rather to present this briefly as a potential future target for rescue therapy in suboptimal DBS cases using the two currently accepted targets-the STN and GPi.

- The lack of originality characterize also the sections devoted to ET and dystonia, where the Authors in an attempt to be all-encompassing create confusion and the reader does not receive adequate information on what to do and what are the best indications for DBS. This is particular evident when they in an attempt to correlate correct targeting and clinical outcome base their thought on old lesional concepts, without any reference to a more actual view of DBS functioning that considers the possibility of an action carried out on neuronal pathways rather than on a single neuronal population.

We have made some corrections to hopefully make it less confusing.  However, again, we were not attempting at any point to specify initial indications for DBS for ET and dystonia outside of a very brief introduction in both sections.  The focus of the paper is on rescue therapies for suboptimal DBS outcomes.  This paper was also meant to be a clinical paper given the requested review. Discussing the intricacies of the underlying mechanisms of DBS which continues to be a mystery was outside the scope of this paper.

- The discussion is very short and does not fit well into the merits of the matter described.

 We appreciate the reviewer’s comments.  The discussion has been expanded.

Reviewer 2 Report

Page

Line

Comment

1

32-33

The sentence "It has indications for several movement   disorders as well as, more recently, psychiatric conditions." can be   deleted given that the conditions in which DBS is indicated for is eventually   repeated later in the paragraph.

1

38

change "which" to "that"

2

56-57

Authors mention: " the patient and clinicians are left with   the difficult decision of what to do next". We strongly urge authors to   include, or at least, wave at the potential ethical issue associated with   "how to obtain patient consent" before doing something. Initial   consent to get the implant in the brain does not automatically mean a   subsequent consent; as some cases were reported and discussed in the   literature. For instance, authors could include a sentence such as   "Obtaining patient consent to undergo postoperative procedure might come   with difficulties and deserve careful attention by the team". Authors do   not need to get into the details, but authors should allude to potential   consent issue by referring to work discussing some of these cases , for   example, 

  Gilbert, F., (2014) Self-Estrangement & Deep Brain Stimulation: Ethical   issues related to Forced Explantation    Neuroethics. 8(2): 107-114 DOI 10.1007/s12152-014-9224-1
   Gilbert, F., (2013) Deep Brain   Stimulation for Treatment Resistant Depression: Postoperative Feeling of   Self-Estrangement, Suicide Attempt and Impulsive-Aggressive Behaviours.   Neuroethics.  Vol 6, Issue 3, 473-481   DOI: 10.1007/s12152-013-9178-8

2

67

Insert a comma after "motor features"

2

67

Insert "features" after "non-motor"

2

69

Might be better to delete the word "also" after   "therapy"

2

70

Delete apostrophe after "years"

2

71

Insert a hyphen between "medication" and   "related"

2

76

Replace "Benefits overall" with "Overall   benefits"

2

76

Replace "out" with "for up"

2

82

Replace "identified which have continued to be   explored." with "identified, which are still continuously being   explored."

2

84

What are these neuropsychiatric symptoms?

2

85

Maybe insert a concluding sentence to this paragraph stating how   surgeons usually determine which region to target given GPi and STN's   seemingly equal benefits. Given that STN stimulation allows for greater   medication reduction post-surgery, is it usually given to patients who   exhibit worse side effects from medications? On the other hand, since GPI   stimulation seemed more advantageous for neuropsychiatric symptoms, is it   then usually prescribed to those who exhibit them? Is there an established   consensus on which region to stimulate depending on PD symptoms, stage, etc.?

2

91

Change "lessoning" to "lessening"

3

114

Did any of these reports indicate why they only chose to   stimulate the STN after the rescue procedure and not perform concurrent STN   and GPi stimulation in these patients?

3

122-143

Not reallly sure if this paragraph should be included in the   review since PPN implantation was not really performed as a rescue procedure   in the cases mentioned.

3

146

Perhaps add a short phrase after "vice versa)."   mentioning that subthalamotomy has also been performed as a rescue procedure

4

147-148

The sentence on PPN stimulation could perhaps be removed given   that the cases mentioned did not really do it as a rescue procedure. If the   authors wish to keep it, they could potentially refer to PPN stimulation as a   potential rescue procedure in the future, citing results from the studies   they have mentioned in lines 122-143.

4

172-178

Did the authors mention why they did not perform concurrent cZI   and VIM stimulation?

4

184

Do the series reported present the results for each individual   patient? If they do, what are the results for the three people that have   essential tremor?

4

184-186

What's the potential effect of the type of tremor to treatment   outcomes? Do the mentioned types of tremor have a different neurobiological   underpinning that could potentially affect the patients' eligibility for VIM   stimulation and/or influence the decision on which region should be implanted   for the rescue stimulation?

4-5

188-203

The paragraphs on unilateral thalamotomy and insertion of a   second electrode in the thalamus could be switched for continuity (mentioning   studies involving implantation in another region first and then mentioning   those that involved thalamotomy afterwards).

4

193-194

Replace "had significant, persistent adverse effects from   thalamotomy after DBS [49]. This was
  reported as facial numbness." with "had a significant persistent   adverse effect in the form of facial numbness from thalamotomy after   DBS."

5

217

Similar to what you did for PD and ET, could you mention some   common treatments or medication prescribed to those with dystonia?

5

241-242

Given that "there is a long latency between programming   adjustments and  resulting clinical   benefit that can be months in duration in dystonia", do you think the   lack of benefit observed in the statement "Notably, stimulation of the   rescue leads by themselves was not able to achieve symptomatic control in   either case." is substantiated? How long did the authors perform the   comparison between concurrent stimulation and just STN stimulation?

5

242-243

Replace "17 months and 15 months, respectively" with   "17 months for the left GPi rescue operation and 15 months for the   bilateral STN rescue procedure."

5-6

244-259

You could perhaps remove these two paragraphs given that the   cases you've mentioned in them were not reallly rescue procedures for   suboptimal DBS results.

6

256-259

As with the previous comment, I think you could remove this   paragraph. However, you could use the results mentioned in lines  256-259 to further support the results of   the case presented in lines 240-243.

6

260-261

Maybe you could emphasize that there are only two reported   patients (based on your review) having dystonia who underwent a rescue   procedure after suboptimal DBS results.

6

266

Replace "who is" with "what qualifies as"

6

268

Maybe you could add "for movement disorders" after   "indications" since your review did not cover rescue procedures for   those who have epilepsy, OCD, or other conditions in which DBS is already   approved for.

6

272

You mentioned that most data on rescue procedures is   "almost entirely in the form of case reports and series"; however,   I think this might be the only source of data on this subject. It might be   unethical to do a randomized clinical trial just on rescue procedures alone,   and it might not also be practically feasible given the number of confounding   variables that have to be considered.

6

267-276

I think you could expand your discussion further by expounding   on some salient issues and concerns. When is it actually necessary to perform   a second procedure rather than just change the settings or medication dosage?   Also, what are the factors that have to be considered in selecting the   appropriate rescue procedure? In procedures involving implantation of leads   in a new region, how do you choose that region? Some cases also indicated   implantation in regions that don't really have regulatory approval for that   condition yet. Finally, are there currently established guidelines and   recommendations for rescue procedures in the conditions that you have   discussed? Perhaps, you could also advocate for the establishment of such.

Author Response

We appreciate the reviewer taking the time to review our manuscript and providing us with such comprehensive comments.  Our responses are attached in a word file.

Round 2

Reviewer 1 Report

It has appreciated the effort of the Authors in correcting and implement their manuscript that has taken the best shape. But we believe that it should be implemented even with a series of considerations and bibliographic references, that they have not considered and that instead in our opinion are of greater importance:

- PPN, though the term used in the literature, can 'be confusing and we prefer to use the term PPTg (nucleus tegmenti pedunculopontini) in accordance with the stereotactic atlas of Paxinos and Huang (1995) and with the cytoarchitectonic atlas of Olzewsky and Baxter .

-  It must be pointed out that often the most widespread literature refers to reviews or case reports or case series quantitatively scarce and should be given  only jobs that relate to larger clinical series please cite and comment :

- Mazzone et al. Ten years of experience using deep brain stimulation: elucidating the mechanism of action of stimulation of the ventrolateral pontine tegmentum.Journal of Neural Transmission, 2016.Special Issue 

- Is necessary to quote, discuss  and analize the extent to which we refer to PPTg the following bibliography references:

1): The clinical effects of deep brain stimulation of the pedunculopontine tegmental nucleus in movement disorders may not be related to the anatomical target, leads location, and setup of electrical stimulation. Mazzone, P., Sposato, S., Insola, A., Scarnati, E. Year the Document was Publish 2013 Source of the Document Neurosurgery 73 (5), pp. 894-906

2) The pedunculopontine tegmental nucleus: Implications for a role in modulating spinal cord motoneuron excitability Authors of Document Scarnati, E., Florio, T., Capozzo, A., Confalone, G., Mazzone, P.2011 Source of the Document.Journal of Neural Transmission

3) The function of the nucleus pedunculopontini in the preparation and execution of an externally-cued task bar pressure in the rat

T Florio, A Capozzo, E Puglielli, Pupil R, G Pizzuti, And Scarnati

Behavioural Brain Research 104 (1), 95-104

4) Cholinergic excitation from the pedunculopontine tegmental nucleus to the dentate nucleus in the rat Authors of Document: Vitale, F., Mattei, C., Capozzo, A., (...), Mazzone, P., Scarnati, E. the Year Document was Publish 2016 Source of the Document: Neuroscience.

5) Commentary: The pedunculopontine nucleus: Clinical experience, basic questions and future directions Authors of Document Mazzone, P., Scarnati, E., Garcia-Rill, E. Year Publish 2011 was the Document Source: Document of the Journal of Neural Transmission 118 (10), pp. 1391-

 its discussion can only add value to the paper of the Authors making it right for the publication.

Author Response

It has appreciated the effort of the Authors in correcting and implement their manuscript that has taken the best shape. But we believe that it should be implemented even with a series of considerations and bibliographic references, that they have not considered and that instead in our opinion are of greater importance:

We appreciate the comments.  We worked hard to address concerns in the prior reviews.

- PPN, though the term used in the literature, can 'be confusing and we prefer to use the term PPTg (nucleus tegmenti pedunculopontini) in accordance with the stereotactic atlas of Paxinos and Huang (1995) and with the cytoarchitectonic atlas of Olzewsky and Baxter .

Given this request, we have now included both terms in several places within the manuscript.

-  It must be pointed out that often the most widespread literature refers to reviews or case reports or case series quantitatively scarce and should be given  only jobs that relate to larger clinical series please cite and comment :

- Mazzone et al. Ten years of experience using deep brain stimulation: elucidating the mechanism of action of stimulation of the ventrolateral pontine tegmentum.Journal of Neural Transmission, 2016.Special Issue 

1): The clinical effects of deep brain stimulation of the pedunculopontine tegmental nucleus in movement disorders may not be related to the anatomical target, leads location, and setup of electrical stimulation. Mazzone, P., Sposato, S., Insola, A., Scarnati, E. Year the Document was Publish 2013 Source of the Document Neurosurgery 73 (5), pp. 894-906

2) The pedunculopontine tegmental nucleus: Implications for a role in modulating spinal cord motoneuron excitability Authors of Document Scarnati, E., Florio, T., Capozzo, A., Confalone, G., Mazzone, P.2011 Source of the Document.Journal of Neural Transmission

3) The function of the nucleus pedunculopontini in the preparation and execution of an externally-cued task bar pressure in the rat

T Florio, A Capozzo, E Puglielli, Pupil R, G Pizzuti, And Scarnati

Behavioural Brain Research 104 (1), 95-104

4) Cholinergic excitation from the pedunculopontine tegmental nucleus to the dentate nucleus in the rat Authors of Document: Vitale, F., Mattei, C., Capozzo, A., (...), Mazzone, P., Scarnati, E. the Year Document was Publish 2016 Source of the Document: Neuroscience.

5) Commentary: The pedunculopontine nucleus: Clinical experience, basic questions and future directions Authors of Document Mazzone, P., Scarnati, E., Garcia-Rill, E. Year Publish 2011 was the Document Source: Document of the Journal of Neural Transmission 118 (10), pp. 1391-

We appreciate the reviewer’s expertise on the PPN.  We have cited several of the requested references from Mazzone et al as indicated above in a summary sentence at the end of the PPN section.  However, as indicated in the prior review round, the purpose of this review manuscript was not to give an exhaustive review of the PPN or PPTg in and of itself.  Rather, we were focused on overall rescue therapies and, in this particular section, the potential of the PPN/PPTg as a rescue therapy.  The other reviewer asked us to cut down the PPN section or perhaps eliminate it.  We agree with reviewer 1 that it should at least be mentioned and did not want to eliminate it.  However, more in depth discussion was outside the scope of this review manuscript on rescue therapies.    

Round 3

Reviewer 1 Report

The reviewer appreciated the effort of the authors in implementing the paper in accordance with the suggestions  and notes proposed for the modification of the previous version of the manuscript .

- the only last  correction that must be made is in the  lines  129 -130:  here have to change in agreement with the correct nomenclature " the pedunculopontine tegmental nucleus", acronym :PPTg.
